# Risk Factors of Initial Inappropriate Antibiotic Therapy and the Impacts on Outcomes of Neonates with Gram-Negative Bacteremia

**DOI:** 10.3390/antibiotics9040203

**Published:** 2020-04-23

**Authors:** Shih-Ming Chu, Jen-Fu Hsu, Mei-Yin Lai, Hsuan-Rong Huang, Ming-Chou Chiang, Ren-Huei Fu, Ming-Horng Tsai

**Affiliations:** 1Division of Pediatric Neonatology, Department of Pediatrics, Chang Gung Memorial Hospital, Taoyuan 333, Taiwan; kz6479@cgmh.org.tw (S.-M.C.); jeff0724@gmail.com (J.-F.H.); lmi818@msn.com (M.-Y.L.); qbonbon@gmail.com (H.-R.H.); cmc123@cgmh.org.tw (M.-C.C.); rkenny@cgmh.org.tw (R.-H.F.); 2Chronic Diseases and Health Promotion Research Center, Chang Gung University of Science and Technology, Chiayi 613, Taiwan; 3Division of Neonatology and Pediatric Hematology/Oncology, Department of Pediatrics, Chang Gung Memorial Hospital, Yunlin 638, Taiwan; 4College of Medicine, Chang Gung University, Taoyuan 333, Taiwan; 5Division of Neonatology and Pediatric Hematology/Oncology, Department of Pediatrics, Yunlin Chang Gung Memorial Hospital, 707, Gongye Rd, Sansheng, Mailiao Township, Yunlin 638, Taiwan

**Keywords:** mortality, antimicrobial therapy, septicemia, late-onset sepsis, antibiotic resistance

## Abstract

Background: Timely appropriate empirical antibiotic plays an important role in critically ill patients with gram-negative bacteremia. However, the relevant data and significant impacts have not been well studied in the neonatal intensive care unit (NICU). Methods: An 8-year (1 January 2007–31 December 2014) cohort study of all NICU patients with gram-negative bacteremia (GNB) in a tertiary-care medical center was performed. Inadequate empirical antibiotic therapy was defined when a patient did not receive any antimicrobial agent to which the causative microorganisms were susceptible within 24 h of blood culture sampling. Neonates with GNB treated with inadequate antibiotics were compared with those who received initial adequate antibiotics. Results: Among 376 episodes of Gram-negative bacteremia, 75 (19.9%) received inadequate empirical antibiotic therapy. The cause of inadequate treatment was mostly due to the pathogen resistance to prescribed antibiotics (88.0%). Bacteremia caused by *Pseudomonas aeruginosa* (Odds ratio [OR]: 20.8, *P* < 0.001) and extended spectrum β-lactamase (ESBL)-producing bacteria (OR: 18.4, *P* < 0.001) had the highest risk of inadequate treatment. Previous exposure with third generation cephalosporin was identified as the only independent risk factor (OR: 2.52, 95% CI: 1.18–5.37, *P* = 0.018). Empirically inadequately treated bacteremias were significantly more likely to have worse outcomes than those with adequate therapy, including a higher risk of major organ damage (20.0% versus 6.6%, *P* < 0.001) and infectious complications (25.3% versus 9.3%, *P* < 0.001), and overall mortality (22.7% versus 11.0%, *P* = 0.013). **Conclusions**: Inadequate empirical antibiotic therapy occurs in one-fifth of Gram-negative bacteremias in the NICU, and is associated with worse outcomes. Additional prospective studies are needed to elucidate the optimal timing and aggressive antibiotic regimen for neonates who are at risk of antibiotic-resistant Gram-negative bacteremia.

## 1. Background

Bloodstream infection is the major cause of mortality in the neonatal intensive care unit (NICU), after more advanced perinatal care and improved delivery room resuscitation have been achieved in the recent years [1,2]. Although coagulase-negative staphylococci remains the most common pathogen of neonatal bacteremia, there is an increasing trend of Gram-negative bacteremia (GNB) in the NICU [3,4], especially after long duration of hospitalization, gram-negative bacteria colonization, or underlying gastrointestinal pathology [5,6,7]. The emergence of antibiotic resistance among GNB is a great concern, because the use of broad-spectrum antibiotics for antibiotic resistant GNB may cause a vicious cycle [8,9,10]. The condition may become especially critical when severe GNB, defined as a fulminant and rapidly devastating sepsis, is encountered.

Appropriate initial antibiotic therapy has been demonstrated as the key independent factor of treatment outcomes in patients with Gram-negative bacteremia in previous studies from the adult ICUs or wards [11,12,13,14]. However, this issue has not been fully studied in the NICU, except for a small sample size, case-control study found in the literature [15]. GNB in the NICU are well known to potentially cause life-threatening septic shock, especially by multidrug resistant (MDR) pathogens or in neonates with underlying comorbidities [4,16]. MDR GNB accounts for nearly one-fifth of all neonatal GNBs, or 7%–10% of all neonatal late-onset sepsis [4,17,18]. Neonates with MDR GNB are well known to have a significantly higher risk of mortality than those with antibiotic-susceptible GNB [4,18]. Besides this, it is worthwhile to examine clinical parameters that can be used as prognostic markers in critically ill neonates, which can facilitate the assessment of the GNB course and guide treatment strategies. Therefore, we aimed to investigate the impact of initial inadequate antibiotic therapy on outcomes of neonates with GNB and examine clinical parameters that can be predictors of final adverse outcomes. 

## 2. Patients and Methods

### 2.1. Setting and Study Design

This study was conducted in the NICUs of Chang Gung Memorial Hospital (CGMH), and all neonates hospitalized in the NICUs of Linkou CGMH were enrolled. These NICUs contains a total of three units and has a total capacity of 47 beds equipped with mechanical ventilator, and 48 beds with special care nurseries. Since 2003, all neonates admitted to our NICUs were included in a database and followed up until death or hospital discharge. This database contained basic demographics, comorbidities of prematurity, all nosocomial infections, and discharge diagnosis. We conducted a retrospective cohort study using this prospectively collected database. Written consent for this study was not required by the Ethics Committee of CGMH, which approved this research. 

All patients between January 2007 and December 2014 with microbiologically documented GNB were enrolled for analysis. Polymicrobial infections were also included if one of the pathogens was a Gram-negative organism. Subsequent episodes of bacteremia in study patients within one month from the first episode of Gram-negative bacteremia were excluded due to better outcome analysis. We reviewed the medical records of the neonates with initial inadequate antibiotics and compared them with those who received appropriate initial antimicrobial therapy. The main outcome measures included the infectious complications, early mortality (death within 7 days of bacteremia onset), and overall mortality (death due to any reason within 30 days of bacteremia onset).

### 2.2. Data Collection

In addition to all basic demographics and complications of prematurity retrieved from our neonatal database, chart and electronic medical records of patients who met the inclusion criteria were reviewed for the presence of the following features: details of clinical features, laboratory data and treatment courses of bacteremias, initial antimicrobial therapy regimen, use of central venous catheter (CVC), total parenteral nutrition (TPN) and/or intrafat, mechanical ventilators, underlying comorbidities, surgical interventions, and receipt of corticosteroid or antibiotics within 30 days prior to bacteremia. Severity of illness was calculated by two investigators (Dr. S.-M.C. and Dr. J.-F.H.) based on the Neonatal Therapeutic Intervention Scoring System (NTISS) [19].

### 2.3. Definitions

Gram-negative bacteremia was defined as the identification of gram-negative bacilli in a blood culture specimen. Only clinically significant bacteremias were enrolled, which was defined as at least one positive blood culture, together with clinical features compatible with systemic inflammatory response syndrome [4,7,12]. The bacteremia onset was defined as the time point of blood culture sampling through peripheral vein or artery, as we never obtained the blood culture from CVC. Inadequate antimicrobial therapy referred to the administration of antimicrobial agents to which the causative microorganisms were resistant in vitro within the first 24 h of bacteremia onset, or lack of administration of an antimicrobial therapy. 

In our institution, blood cultures were incubated in the Bactec 9240 system, and antibiotic susceptibility testing was performed by using the disk diffusion method according to the recommendations of the Clinical Laboratory Standards Institute [20]. Multidrug resistance was defined using previously published criteria [21]. In our NICU, empirical antibiotics were prescribed for the coverage of both Gram-positive and Gram-negative organisms, usually oxacillin or vancomycin plus cefotaxime or gentamicin, once late-onset sepsis was suspected. For early-onset sepsis, ampicillin plus gentamicin or cefotaxime were usually prescribed, depending on the physician’s decision. The microbiology laboratory would notify the clinician when a blood culture was positive, and the clinician was responsible to modify the antimicrobial regimens according to subsequent bacterial identification and antimicrobial susceptibility results.

Early-onset sepsis and late-onset sepsis were defined as clinical sepsis combined with bacterial growth on a blood culture obtained within the first 72 h of life and > 72 h of life, respectively [5,8]. A bacteremia was classified as community-acquired in this study if the neonate had been discharged from NICU or baby room for at least two days, and the first positive blood culture occurred ≤ 48 h after hospital admission. Antimicrobial exposure was defined as systemic administration of an antibiotic class for at least 3 days in the preceding 30 days before the onset of bacteremia. All comorbidities of prematurity, including respiratory distress syndrome (RDS), intraventricular hemorrhage (IVH), bronchopulmonary dysplasia (BPD), necrotizing enterocolitis (NEC), and periventricular leukomalacia (PVL) were based on the latest updated diagnostic criteria in the standard textbook of neonatology [22]. Shock was defined as a mean blood pressure < lower limit according to gestational age that was unresponsive to fluid treatment or required vasoactive agents [23]. Persistent bacteremia was defined as two or more consecutive positive blood cultures for more than 48 h on appropriate antibiotic therapy. Infectious complications were defined as any newly infectious focus or persistent organ damage that occurred within one week after the onset of bacteremia, but not concurrently.

### 2.4. Statistical Analysis

The student’s *t* test or Mann–Whitney *U*–test were used to compare continuous variables depending on values distribution, and χ^2^ or Fisher’s exact test was used to compare categorical variables. A two-sided *P* value of < 0.05 was considered significant. In identifying the independent risk factors for mortality, a backward stepwise logistic regression analysis was used to control for the effects of confounding factors. Variables with a *P* value of < 0.05 in the univariate analysis were candidates for multivariate analysis as well as the main variable of interest (i.e., inappropriate initial antimicrobial therapy). Kaplan–Meier analysis was used to compare the survival situation of neonates with GNB treated with appropriate versus inappropriate antibiotics. Statistical analyses were performed using SPSS version 15.0 (SPSS^®^, Chicago, IL).

### 2.5. Availability of Data and Materials

The datasets used/or analyzed during the current study are available from the corresponding author on reasonable request.

### 2.6. Ethics Approval and Consent to Participate

This study was approved by the institutional review board of Chang Gung Memorial Hospital, (the IRB certificate number: 103-3236B) with a waiver of informed consent because all patient records and information were anonymized and de-identified prior to analysis.

## 3. Results

### 3.1. The Epidemiology of Gram-Negative Bacteremia in the NICU

During the study period, a total of 333 neonates with 395 episodes of Gram-negative bacteremia were identified; 19 of them were the recurrent episodes which occurred within one month after the first episode and were excluded, leaving a total of 333 neonates with 376 episodes of Gram-negative bacteremia for analysis (Table 1).

Most of our cases were nosocomial infections (371 episodes, 98.7%), and only 5 episodes (1.3%) were community-acquired. There were 354 episodes of late-onset sepsis and 22 episodes of early-onset sepsis. The predominant organisms for monomicrobial bacteremias were *Klebsiella* spp. (n = 122; 32.4%), *Escherichia coli* (n = 92; 24.5%), *Enterobactor* spp. (43; 11.4%), and *Acinetobacter baumannii* (43; 11.4%). Thirty-three episodes of bacteremias were polymicrobial (8.8%). There were 70 (18.6%) multidrug-resistant organisms among the isolates, including 47 extended-spectrum β-lactamase (ESBL) producers.

### 3.2. The Frequency of Inadequate Antibiotic Treatment of Gram-Negative Bacteremia

The antibiotics with Gram-negative activity that were most frequently prescribed during the first 24 h of bacteremias onset were cefotaxime (228; in 60.6% of episodes), gentamicin (51; 13.6%), meropenem (46; 12.2%), ceftazidime (38; 10.1%), and aztreonam (4; 1.1%). In 9 (2.4%) episodes, only monotherapy with vancomycin was prescribed initially. In 348 cases (92.6%), more than one antibiotic were used during this time period.

Seventy-five (19.9%) episodes of Gram-negative bacteremia were treated with inadequate empirical antibiotics. In 66 (88.0%) of these cases, subsequently identified pathogens resistant to the prescribed antibiotics were responsible for the inadequate treatment, including multidrug resistant strains for 51 (68.0%) isolates. For the other 9 (12.0%) cases, the clinicians failed to administer in-vitro susceptible antibiotics within 24 h after the initial positive blood culture was drawn. The median time from bacteremia onset to receive adequate antibiotic treatment in those without initially adequate treatment was 47 h (range: 29–94 h). Within 24 h after notification of antibiotic susceptibility, all patients received adequate antibiotic treatment.

*Klebsiella pneumoniae* (n = 15, 20.0%) was the most common pathogen treated with initially inadequate antibiotic, followed by *E. coli* (14, 18.7%) and *Pseudomonas aeruginosa* (12, 16.0%). However, ESBL-producing bacteria (n = 34, 72.3%, odds ratio [OR]: 18.4, 95% confidence interval [CI]: 9.0–37.7, *P* < 0.001) and *Pseudomonas aeruginosa* (n = 12, 75.0%, OR: 20.8, 95% CI: 5.8–75.3, *P* < 0.001) had the highest rate of receiving inadequate antibiotics initially (Table 1).

### 3.3. Factors Associated with Inadequate Empirical Antibiotic Treatment of Gram-Negative Bacteremia

Between patients receiving initially inadequate and adequate empirical treatment, there were no significant differences in terms of demographics, age at onset of bacteremia, use of CVC, TPN and/or intrafat, mechanical ventilation, and underlying chronic conditions (Table 2). Initially inadequately treated bacteremias were more likely to be the recurrent episode (37.3% vs. 22.9%, *P* = 0.018), and significantly more likely to have antibiotic exposure with a third generation cephalosporin (*P* < 0.001) and vancomycin (*P* = 0.004) within one month before bacteremia onset than adequately treated bacteremias. In terms of clinical manifestations and laboratory data at onset of bacteremia, there were no significant differences between the two groups, except higher serum C-reactive protein concentration in the bacteremias without adequate treatment (median, 78.0 vs. 59.1, *P* = 0.038). With multivariate logistic regression analysis, only previous antibiotic exposure to a third generation cephalosporin within one month was identified as the independent risk factor (Odds ratio [OR]: 2.52, 95% CI: 1.18–5.37, *P* = 0.018). 

### 3.4. The Outcome of Inadequate Empirically Treated Gram-Negative Bacteremia

Table 3 shows the comparison of clinical outcomes between the patients with and without initially adequately treated bacteremias. Although the rates of persistent bacteremia were comparable, significantly more patients with inadequately treated bacteremias had prolonged feeding intolerance (> 3 days) and higher severity of illness at the third day of bacteremia (scored by NTISS). Besides this, infants with Gram-negative bacteremias but without initially adequate antibiotic therapy had a significantly higher risk of progression to severe sepsis or septic shock (*P* = 0.002) and infectious complications (*p* < 0.001). 

For bacteremias without initial adequate therapy, a total of nine patients died within 7 days, including 5 patients who died within 48 h after bacteremia onset without receiving any effective antimicrobial therapy. Infants without initially adequate antibiotic therapy did not have a significantly higher early mortality rate than those with adequately therapy, and the risk of recurrent bacteremia within one month was also comparable (Table 3). However, inadequately treated bacteremias had significantly higher overall mortality rate than adequately treated bacteremias (22.7% vs. 11.0%, *P* = 0.013). This can also be confirmed by log rank test (p = 0.051) [Figure 1].

## 4. Discussion

Results from this study indicated that initially inadequate antimicrobial treatment was significantly associated with antibiotic resistant Gram-negative bacteria, mainly *P. aeruginosa* and ESBL-producing bacteria. Previous antibiotic exposure to a third generation cephalosporin was identified as the independent risk factor for initially inadequately treated bacteremias before blood culture results and antibiotic susceptibilities were available. The possible explanation could be that broad-spectrum cephalosporin exposure caused selection of drug-resistant bacteria, which further resulted in inadequately treated bacteremias, if alternatively more effective antibiotics were not prescribed. The adverse outcomes were in patients with the significantly higher rates of infectious complications and overall mortality (within 30 days) compared with adequately treated patients.

Several factors have been found for inadequate treatment in other settings, e.g., hospital admission in the 90 days prior to the current admission, admission to surgery, nosocomial infection, and polymicrobial infection [16,24,25]. All these factors were not found in the present study. Our finding was consistent with previous reports that previous antibiotic exposure to broad-spectrum cephalosporin is associated with a significantly higher risk of infection caused by antibiotic-resistant bacteria [26,27,28,29]. Although some other broad-spectrum antibiotics, such as carbapenem or fluoroquinolones, are also found to cause antibiotic resistant Gram-negative bacteremia [28,30], these antibiotics were much less frequently prescribed in our cohort. Sometimes, bacteremias were inadequately treated because antibiotic-resistant bacteremias are indistinguishable from those that are antibiotic-susceptible. In addition, the initial partial response of these inadequately treated bacteremias would delay clinicians’ decision to modify antibiotics, and sometimes clinicians may have ignored the clinically worsening progression.

The appropriateness of initial antibiotics depends on the empirical antibiotic prescribed and the local antibiotic susceptibility patterns. It seemed reasonable that the lower the rate of drug resistant organisms in the NICU, the lower the rate of initially inadequate antimicrobial therapy. Therefore, the issue of how to reduce antimicrobial resistance in the ICU [31,32] is important but not fully studied in the neonatal settings. For example, in contrast to previous studies indicating that *S. marcescens* and *A. baumannii* causing bacteremias were multidrug resistant and associated with high mortality or morbidity in the NICU [33,34], all of the *S. marcescens* and most of the *A. baumannii* isolates in our cohort were susceptible to aminoglycoside or cefotaxime, and thus, they were not difficult to treat. The significantly higher mortality of inadequately treated Gram-negative bacteremias may partially be explained by the pathogens themselves, and *P. aeruginosa* and ESBL-producing bacteria, the major causative organisms of this study, have been found to be more likely to cause fulminant illness or early mortality [35,36]. 

In this study, the result in neonates with GNB is comparable with that documented in the literature of studies which were conducted in adults [15,24,25]. Gram-negative bacteremias without initially adequate antimicrobial therapy had a significantly higher risk of overall mortality, prolonged illness, progression to severe sepsis, and infectious complications, but not early mortality within seven days of onset. It seemed that in neonates, Gram-negative bacteremias without initially adequate antimicrobial therapy did not cause early mortality within seven days of onset but they caused a prolonged illness, major organs damage, infectious complications, and then led to clinical deterioration, which subsequently resulted in final in-hospital mortality. 

This study highlighted the importance of further efforts to elucidate the optimal timing and aggressive antibiotic regimen for neonates who are at risk of antibiotic-resistant Gram-negative bacteremia. Antimicrobial stewardship interventions have been well studied to prompt appropriate antibiotics prescription in the NICU [37,38,39]. In our institute, routine antifungal prophylaxis for extremely preterm neonates has been launched since early 2017, and the influences as well as those of antimicrobial stewardship intervention are currently under investigation. Our previous studies found that MDR GNB accounted for 18.6% of all neonatal GNB [4,7,16].

Because colonization of the newborns with MDR Gram-negative pathogens makes them the potential source of nosocomial outbreaks [8,40], further routine surveillance of colonization is also suggested.

There are some limitations in this study. This study was observational and not randomized; therefore, the choices of initial empirical antibiotic depended on attending physicians, instead of the researchers. We failed to account for some important confounders that might influence the decision of attending physician in modifying antibiotics, such as the clinical progression on the first day of bacteremia. Besides this, the data were derived from a single center; therefore our conclusions are necessarily limited in generalizability to other settings and institutions. 

## 5. Conclusions

In conclusion, initial inappropriate antibiotic therapy in neonates with gram-negative bacteremia is associated with a significant higher risk of overall mortality and infectious complications. Early identification of neonatal sepsis caused by antibiotic resistant pathogens, such as ESBL producing gram-negative bacilli or *Pseudomonas* spp, may reduce the rate of inadequate antibiotic treatment, which can potentially improve the treatment outcomes. 

## Figures and Tables

**Figure 1 antibiotics-09-00203-f001:**
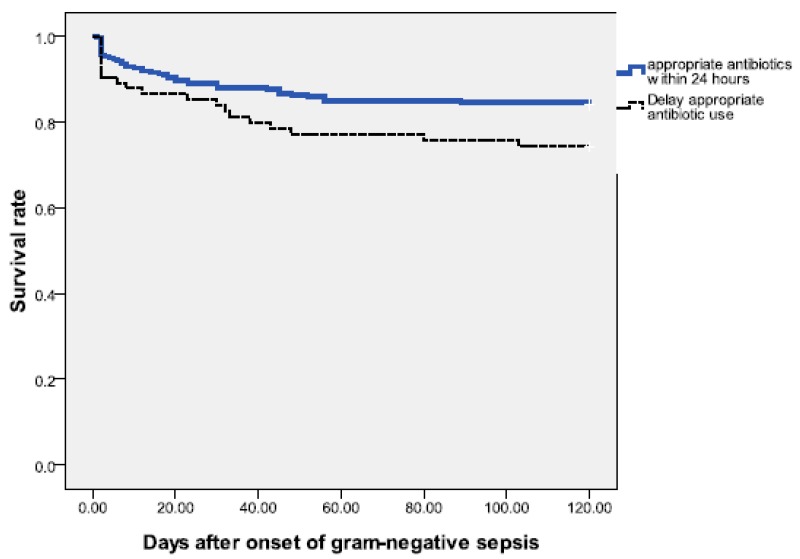
Survival following onset of gram-negative sepsis from neonates in the neonatal intensive care unit (NICU). The Kaplan–Meier graph is stratified by the episodes treated with appropriate antibiotics within 24 h and those with delayed initial appropriate antibiotic treatment; [log rank test = 0.051].

**Table 1 antibiotics-09-00203-t001:** Bacterial isolates in 333 neonates with 376 episodes of Gram-negative bacteremia.

Microorganism	Total Case No. (%)	Inadequate Treatment, n (% of Each Type of Bacteria)
Early-onset sepsis	22 (5.9)	
*Escherichia coli*	16 (4.3)	2 (12.5)
Others	6 (1.6)	1 (16.7)
Late-onset sepsis	354 (94.1)	
*Klebsiella pneumoniae*	82 (21.8)	15 (18.3)
*Escherichia coli*	76 (20.2)	12 (15.8)
*Acinetobacter baumannii*	43 (11.4)	4 (9.3)
*Klebsiella oxytoca*	39 (10.4)	5 (12.8)
*Enterobacter cloacae*	26 (6.9)	8 (30.8)
*Enterobacter aerogenes*	17 (4.5)	4 (23.5)
*Pseudomonas aeruginosa*	16 (4.3)	12 (75.0)
*Serratia marcescens*	10 (2.7)	0 (0)
Others *	12 (3.2)	4 (33.3)
Polymicrobial organisms ^¶^	33 (8.8)	8 (24.2)
Total	376 (100)	75 (19.9)
ESBL producing bacteria ^&^	47 (12.5)	34 (72.3)

* Including Citrobacter freundii (3), Stenotrophomonas maltophilia (3), Hafnia alvei (2), Neisseria Meningitidis (2), Chryseobacterium meningoseptium (1) and Flavobacterium (1). ^¶^ Indicating two or more microorganisms were recovered from the same blood culture set; ^&^ 47 ESBL (extended-spectrum β-lactamase) producing bacteria were included in the total 376 episodes, and included *K. pneumonia* (28), *E. coli* (9), *K. oxytoca* (6), and *E. cloacae* (4).

**Table 2 antibiotics-09-00203-t002:** Comparison of 376 episodes of Gram-negative bacteremias treated with inadequate versus adequate empirical antibiotic treatment.

	Case No. (%) (Total n = 376)	Univariate Analysis			Multivariate Analysis Odds Ratio (95% CI)
		Inadequate Treatment (n = 75)	Adequate Treatment (n = 301)	*P* Value	
Gestational age (weeks), median (IQR)	30.0 (27.0–35.0)	31.0 (27.0–36.0)	30.0 (27.0–35.0)	0.192	-
Birth body weight (g), median (IQR)	1345.0 (900.0–2051.3)	1600.0 (960.0–2135.0)	1265.0 (895.0–2020.0)	0.084	-
Male gender	191 (50.8)	35 (46.7)	156 (51.8)	0.518	-
Day of bacteremia onset, median (IQR)	25.0 (13.0–54.0)	31.0 (13.0–66.0)	24.0 (13.3–52.8)	0.136	-
Late-onset sepsis	354 (94.1)	72 (96.0)	282 (93.7)	0.854	-
Episode sequence of bacteremia				0.018	
1st episode	279 (74.2)	47 (62.7)	232 (77.1)		1 (reference)
Recurrent episode	97 (25.8)	28 (37.3)	69 (22.9)		1.41 (0.78–2.58)
Underlying chronic conditions					
Congenital anomalies	24 (6.4)	5 (6.6)	19 (6.3)	1.000	-
Neurological sequelaes	57 (15.2)	12 (16.0)	45 (15.0)	0.858	-
Bronchopulmonary dysplasia	170 (45.2)	31 (41.3)	139 (46.1)	0.517	-
Chronic gastrointestinal pathology	17 (4.5)	4 (5.3)	13 (4.3)	0.710	-
Others	23 (6.1)	6 (8.0)	17 (5.6)	0.427	-
Use of central venous catheter	285 (75.8)	59 (78.7)	226 (75.1)	0.551	-
On high frequency oscillatory ventilator	37 (9.8)	7 (9.3)	30 (10.0)	1.000	-
Under invasive ventilation (intubation)	182 (48.4)	35 (46.7)	147 (48.8)	0.797	-
Use of total parenteral nutrition/intrafat	263 (69.9)	56 (74.7)	207 (68.8)	0.398	-
Previous operation (within one month)	56 (14.9)	10 (13.3)	46 (15.3)	0.858	-
Use of steroid (within one month)	13 (34.5)	5 (6.6)	8 (2.7)	0.147	-
Antibiotic exposure (within one month)					
3rd generation cephalosporin	161 (42.8)	47 (62.7)	114 (37.9)	< 0.001	2.52 (1.19–5.37)
Vancomycin or teicoplanin	149 (39.6)	41 (54.7)	108 (35.9)	0.004	0.96 (0.45–2.06)
Carbapenem	22 (5.9)	6 (8.0)	16 (5.3)	0.409	-
Antifungal treatment	12 (3.2)	5 (6.7)	7 (2.3)	0.069	-
Clinical manifestations at bacteremia onset*					
Sepsis-induced hypotension	96 (25.5)	22 (29.3)	74 (24.6)	0.371	-
GI bleeding and/or coagulopathy	147 (39.1)	29 (38.6)	118 (39.2)	1.000	-
Disseminated intravascular coagulopathy	67 (17.8)	15 (20.0)	52 (17.3)	0.495	-
NTISS score, median (IQR)	17.0 (13.0–20.0)	17.0 (12.0–20.0)	17.0 (14.0–20.0)	0.455	-
Laboratory data at onset of bacteremia					
Leukopenia (WBC count < 4000/uL)	94 (25.0)	16 (21.3)	78 (25.9)	0.459	-
Leukocytosis (WBC count > 20,000/uL)	116 (30.9)	28 (37.3)	88 (29.2)	0.208	-
WBC shift to left **	90 (23.9)	21 (28.0)	69 (22.9)	0.366	-
Anemia (hemoglobin < 11.0 mg/dL)	180 (47.9)	39 (52.0)	141 (46.8)	0.441	-
Thrombocytopenia (platelet < 80,000/uL)	189 (50.3)	41 (54.7)	148 (49.2)	0.439	-
C-reactive protein (mg/dL), median (IQR)	60.4 (31.9–112)	78.0 (38.8–136)	59.1 (28.0–103)	0.038	-
Metabolic acidosis	111 (29.5)	24 (32.0)	87 (28.9)	0.671	-

* Within the first 24 h after blood culture sampling, ** Indicating immature WBC ≥ 20% of total WBC. WBC: white blood cell, NTISS: Neonatal therapeutic intervention scoring system, IQR: interquartile range, 95% CI: 95% confidence interval.

**Table 3 antibiotics-09-00203-t003:** Outcome comparisons of Gram-negative bacteremia treated with inadequate antibiotics versus adequate antibiotics.

	Inadequate Treatment (n = 75)	Adequate Treatment (n = 301)	*P* Value
Persistent bacteremia*	3 (4.0)	6 (2.0)	0.309
Prolonged ileus and/or feeding intolerance (> 3 days)	30 (40.0)	53 (17.6)	< 0.001
Progression to septic shock or severe sepsis^¶^	14 (18.7)	16 (5.3)	0.002
NTISS scores at the third day of bacteremia, median (IQR)	17.0 (12.0–20.0)	15.5 (12.0–18.0)	0.010
Infectious complications^#^	19 (25.3)	28 (9.3)	< 0.001
Major organ damage	15 (20.0)	20 (6.6)	0.001
Newly infectious focus	8 (10.7)	13 (4.3)	0.046
Early mortality (within 7 days)	9 (12.0)	23 (7.6)	0.250
Overall mortality (within 30 days due to any reason)	17 (22.7)	33 (11.0)	0.013
Recurrent bacteremia within one month	5 (6.7)	33 (11.0)	0.390

All data were expressed as number (percentage %), unless indicated otherwise. *Defined as two or more consecutive positive blood cultures with at least 48 h apart. ^#^Defined as newly infectious focus or major organ damage after bacteremias, but not occurred concomitantly. Some episodes had both newly infectious focus and major organ damage. ^¶^Indicating bacteremias without septic shock as the initial presentation, but that progressed to septic shock or severe sepsis 24 h later. NTISS: Neonatal Therapeutic Intervention Scoring System, IQR: interquartile range.

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
