# Peer review of "Risk Factors of Initial Inappropriate Antibiotic Therapy and the Impacts on Outcomes of Neonates with Gram-Negative Bacteremia"

_antibiotics, 2020, doi:10.3390/antibiotics9040203_

Round 1
Reviewer 1 Report
Summary
- “Initial inappropriate antibiotic therapy in neonates with gram-negative bacteremia is 24 significantly associated with worse outcomes. Further effort to decrease emergence of antibiotic resistance and highly suspicion of infection by drug-resistant bacteria clinically is important” In my opinion, this does not reflect the article’s main point
Abstract
- The methods section does not describe the statistical methods
- Results: “including more prolonged illness” please provide numbers, eg mean hospitalisation
- R 45 “further effort to – inadequacy” this implies a causal link of inappropriate therapy and mortality, but this is not supported by the article (merely an association)
Background:
- The references for mortality (11-14) are in adults. It is not clear from the text, the reader may think previous research was in non-ICU neonates
Methods
- Subsequent episodes of bacteremia in study patients within one 78 month from the first episode of Gram-negative bacteremia were excluded due to better outcome analysis. What do the authors mean by “dure to better outcome analysis.
- Clinical features compatible with systemic inflammatory response syndrome. how was this defined?
- Statistics: with this method, confounding (especially unmeasured confounders) may still be an issue, as many different factors may influence empirical treatment and mortality. Propensity score matching may be a better method.
- Some patients were included several times, how was this handled statistically.
Results
- 148: what is the local guideline for empiric antibiotic therapy? It would be could to describe this, as it may help the reader to understand why these antibiotics were choses and whether it was standard (according to local guideline) or that there may have been reason to choose a different empirical treatment.
- 121: “Persistent bacteremia was defined as two or more consecutive positive blood cultures 121 with at least 48 hours apart”
- 157. within 24 hours of the initial positive blood culture à within 24 hours of the initial positive blood culture withdrawal.
- “treatment in those without initially adequate 158 treatment was 47 hours (range: 29-94 hours)” Within 24 hours after notification of antibiotic 159 susceptibility, all patients received adequate antibiotic treatment. The delay seems quite long, and relevant for mortality. Please explain.
- 75%, --> 75.0?
- 161-166 : this is all in univariate analyses.
- Table 1 may need better lay out. The esbl refers to all patients, but all other data are given for late versus early onset specifically.
- “Initially inadequately treated bacteremias were more likely to be the recurrent episode (37.3% vs. 22.9%, P = 177 0.018). What is meant by “the recurrent episode?”
- 193: please define infectious complications in methods section
- I am missing the table with the univariate and multivariate analysis for outcome. Early mortality in the inadequate treatment group was “only” 7 patients. With these numbers it is not easy to perform a regression analysis, only a limited number of variables can be included in the model. But according to the title it is one of the aims of the study
Author Response
RE: Antibiotics-753672
Risk factors of initial inappropriate antibiotic therapy and the impacts on outcomes of neonates with gram-negative bacteremia
Dear Editor,
Thank you for your appreciated comments on our manuscript. We had the manuscript revised, all according to the reviewers’ and editor’s suggestions. We underline every change and highlight in red color on the revised manuscript. The replies for the reviewers’ criticisms are as followings. We hope this revised version can be acceptable.
Best regards,
Ming-Horng Tsai
Chief, Division of Neonatology and Pediatric Hematology/Oncology, Department of Pediatrics, Yunlin Chang Gung Memorial Hospital, Taiwan, R.O.C.
Comments from Reviewer No.1 :
“Initial inappropriate antibiotic therapy in neonates with gram-negative bacteremia is significantly associated with worse outcomes. Further effort to decrease emergence of antibiotic resistance and highly suspicion of infection by drug-resistant bacteria clinically is important” In my opinion, this does not reflect the article’s main point
Reply:
Thank you for your instructive advice. I will revise the conclusion of the abstract as “Additional prospective studies are needed to elucidate the optimal timing and aggressive antibiotic regimen for neonates who are at risk of antibiotic-resistant Gram-negative bacteremia.” in the last sentence of the abstract
Abstract
• The methods section does not describe the statistical methods
• Results: “including more prolonged illness” please provide numbers, eg mean hospitalization
• R 45 “further effort to – inadequacy” this implies a causal link of inappropriate therapy and mortality, but this is not supported by the article (merely an association)
Reply:
Thank you for your instructive advice.
I will add statistical methods in the methods section of the abstract as following: Neonates with GNB treated with inadequate antibiotics were compared with those who received initial adequate antibiotics.
For results, I will revise as a higher risk of major organ damage (20.0% versus 6.6%, P < 0.001)
For the final conclusion, I will revise the conclusion of the abstract as “Additional prospective studies are needed to elucidate the optimal timing and aggressive antibiotic regimen for neonates who are at risk of antibiotic-resistant Gram-negative bacteremia.” in the last sentence of the abstract
Background:
• The references for mortality (11-14) are in adults. It is not clear from the text, the reader may think previous research was in non-ICU neonates
Reply:
Thank you for your instructive advice. I will mention this issue and make it clear that these studies were conducted in adult settings, thank you.
I add “previous studies from the adult ICUs or wards” before reference (11-14)
Methods
• Subsequent episodes of bacteremia in study patients within one 78 month from the first episode of Gram-negative bacteremia were excluded due to better outcome analysis. What do the authors mean by “due to better outcome analysis.
Reply:
Thank you for your question. Because the 7-day mortality and 30-day all cause mortality after the GNB were compared between neonates treated with appropriate and those treated with inappropriate antibiotics, subsequent episodes of bacteremia that happened within one month from the previous one in study patients would cause difficulties in considering event-attributable mortality if these episodes were not excluded.
• Clinical features compatible with systemic inflammatory response syndrome. How was this defined?
Reply:
Thank you for your question. I will add citation for the definition of bacteremia that had clinical features compatible with systemic inflammatory response syndrome in the text as [4,7,12]. This will shorten the paragraph of the method section.
• Statistics: with this method, confounding (especially unmeasured confounders) may still be an issue, as many different factors may influence empirical treatment and mortality. Propensity score matching may be a better method.
Reply:
Thank you for your instructive advice. However, we only compared neonates with GNB treated with inappropriate versus appropriate antibiotics, it may not be necessary to use propensity score matching.
• Some patients were included several times, how was this handled statistically.
Reply:
Thank you for your question. Because we considered each episode of GNB as an independent event, all the demographics and related information were based on the situation when the episode of GNB happened. When we discussed the issue of the impact of inappropriate antibiotics on the outcomes, the end point would be event-attributable mortality instead of final mortality of each patient. Therefore, 7-day mortality and 30-day all cause mortality rates were compared between these two groups. Although some demographics of patients with more than two episodes of GNB were considered twice, these would not cause statistical problems.
Results
• 148: what is the local guideline for empiric antibiotic therapy? It would be could to describe this, as it may help the reader to understand why these antibiotics were choose and whether it was standard (according to local guideline) or that there may have been reason to choose a different empirical treatment.
Reply:
Thank you for your instructive advice. I will provide information regarding the local guideline for empiric antibiotic therapy that were used in our NICU in the definition section of the method sections, as following: (page 8, line 151-155)
In our NICU, empirical antibiotics were prescribed for the coverage of both Gram-positive and Gram-negative organisms, usually oxacillin or vancomycin plus cefotaxime or gentamicin, once late-onset sepsis was suspected. For early-onset sepsis, ampicillin plus gentamicin or cefotaxime were usually prescribed, depending on the physician’s decision.
• 121: “Persistent bacteremia was defined as two or more consecutive positive blood cultures with at least 48 hours apart”
Reply:
Thank you for your instructive advice. I will revise the definition of persistent bacteremia as Persistent bacteremia was defined as two or more consecutive positive blood cultures for more than 48 hours on appropriate antibiotic therapy.
• 157. within 24 hours of the initial positive blood culture à within 24 hours of the initial positive blood culture withdrawal.
Reply:
Thank you for your instructive advice. I will revise this sentence as “For other 9 (12.0%) cases, the clinicians failed to administer in-vitro susceptible antibiotics within 24 hours after the initial positive blood culture was drawn.”
• “treatment in those without initially adequate treatment was 47 hours (range: 29-94 hours)” Within 24 hours after notification of antibiotic susceptibility, all patients received adequate antibiotic treatment. The delay seems quite long, and relevant for mortality. Please explain.
Reply:
Thank you for your question. Because it took an average of 24-72 hours to have the results of pathogen and antibiotic susceptibility during the study period, the treatment of those without initially adequate treatment would be delayed so long time. Therefore, initial inadequate antibiotic therapy is suspected to have contributed to infectious complications and 30-day all cause mortality.
• 75%, --> 75.0?
Reply:
Thank you for your instructive advice. I will revise it as 75.0%, thank you.
• 161-166 : this is all in univariate analyses.
Reply:
Thank you for your question. Yes, this is all in univariate analyses.
• Table 1 may need better lay out. The esbl refers to all patients, but all other data are given for late versus early onset specifically.
Reply:
Thank you for your question. All data are given for each type of bacteria, and the percentages are given for all patients, not for late or early onset specifically, thank you.
• “Initially inadequately treated bacteremias were more likely to be the recurrent episode (37.3% vs. 22.9%, P = 0.018). What is meant by “the recurrent episode?”
Reply:
Thank you for your question. This meant that the initially inadequately treated GNB were more likely to be the 2nd or 3rd episode of the late-onset sepsis that the neonate had during his or her hospitalization, than the appropriately treated GNB, which were more likely to be the first episode. It also indicated that the 2nd or 3rd episode of the late-onset sepsis were more likely to be inadequately treated.
• 193: please define infectious complications in methods section
Reply:
Thank you for your instructive advice. The infectious complication has been defined in the last sentence of the Definition section of methods section as (line 122-124) “Infectious complications were defined as any newly infectious focus or persistent organ damage occurred within one week after the onset of bacteremia, but not concurrently.”
• I am missing the table with the univariate and multivariate analysis for outcome. Early mortality in the inadequate treatment group was “only” 7 patients. With these numbers it is not easy to perform a regression analysis, only a limited number of variables can be included in the model. But according to the title it is one of the aims of the study
Reply:
Thank you for your instructive advice.
The title is “Risk factors of initial inappropriate antibiotic therapy and the impacts on outcomes of neonates with gram-negative bacteremia”. Therefore, the aims of this study are
Why some patients with GNB received inappropriate antibiotics?
What is the impact of inappropriate antibiotics on the outcomes of neonates with GNB
It is not difficult to have a table with the univariate and multivariate analysis for outcome, but it will duplicate the result of my previous study in Pediatrics 2014;133(2):e322-9. (reference no. 4, that I am the first author). Although the study periods were different between these two studies, I think the results will be the same. I suggested not having this issue in this manuscript, thank you.

Reviewer 2 Report
This is a very well designed and written study. I have only few minor suggestions for improvement.
- Abstract (line 36): please remove "and'; start new sentence with "Pseudomonas aeruginosa".
- Background (line 53): "limited development of new antimicrobials". This sentence is outdated since many new antimicrobials have been available since 2015.
- Line 61: Delete "chronic" since the study includes only neonates.
- Line 64: Delete "severe" since it is redundant.
- Line 110: Correct "early-onset".
- Statistical analysis: Add Kaplan-Meier analysis to analyze mortality since this was already done and mentioned in the results.
- Results (line 191): Define NTISS.
- Discussion (lines 246-252): It is worth mentioning that these results in neonates are comparable with the literature in adults with appropriate citations.
Author Response
RE: Antibiotics-753672
Risk factors of initial inappropriate antibiotic therapy and the impacts on outcomes of neonates with gram-negative bacteremia
Dear Editor,
Thank you for your appreciated comments on our manuscript. We had the manuscript revised, all according to the reviewers’ and editor’s suggestions. We underline every change and highlight in red color on the revised manuscript. The replies for the reviewers’ criticisms are as followings. We hope this revised version can be acceptable.
Best regards,
Ming-Horng Tsai
Chief, Division of Neonatology and Pediatric Hematology/Oncology, Department of Pediatrics, Yunlin Chang Gung Memorial Hospital, Taiwan, R.O.C.
Comments from Reviewer No.2 :

Abstract (line 36): please remove "and'; start new sentence with "Pseudomonas aeruginosa".
Reply:
Thank you for your instructive advice. I will remove “and”; start new sentence with “Pseudomonas aeruginosa” accordingly.
Background (line 53): "limited development of new antimicrobials". This sentence is outdated since many new antimicrobials have been available since 2015
Reply:
Thank you for your instructive advice. I will revise this sentence as following: because use of broad-spectrum antibiotics for antibiotic resistant GNB may cause a vicious cycle. Besides, I also revise the reference no. 8 and no. 9 for this sentence, thank you.
Line 61: Delete "chronic" since the study includes only neonates.
Reply:
Thank you for your instructive advice. I will delete “chronic” accordingly. I will also use comorbidities in the sentence in the method section, thank you.
Line 64: Delete "severe" since it is redundant.
Reply:
Thank you for your instructive advice. I will delete “severe” accordingly.
Line 110: Correct "early-onset".
Reply:
Thank you for your instructive advice. I will correct “early-onset” in line 110 accordingly.
Statistical analysis: Add Kaplan-Meier analysis to analyze mortality since this was already done and mentioned in the results.
Reply:
Thank you for your constructive advice. I will add Kaplan-Meier analysis to analyze mortality in the statistical analysis section, thank you. In the Statistical analysis as following: Kaplan-Meier analysis was used to compare the survival situation of neonates with GNB treated with appropriate versus inappropriate antibiotics.
Results (line 191): Define NTISS.
Reply:
Thank you for your instructive advice. NTISS has been defined in the data collection of the Patients and methods section, in line 93 with citation [19] as following: Severity of illness was calculated by two investigators (Dr. S.-M.C. and Dr. J.-F.H.) based on the Neonatal Therapeutic Intervention Scoring System (NTISS) [19].
Discussion (lines 246-252): It is worth mentioning that these results in neonates are comparable with the literature in adults with appropriate citations.
Reply:
Thank you for your instructive advice. I will mention this issue that these results in neonates are comparable with the literature in adults with appropriate citations in the discussion section, in line 246 as following, thank you.
In this study, the result in neonates with GNB is comparable with that documented in the literature which were conducted in adults [15,24,25].

Reviewer 3 Report
Dear Authors,
I read with interest the study. Overall it is an interesting study with important clinical implications that will serve as the baseline for the discussion for appropriate antimicrobial stewardship programs in Taiwan and also infection prevention and control measures to reduce the MDR gram-negative bacteria.
My comments can be found below.
1) I am concerned regarding the timeline of the study since the data are until Dec 2014. Since we are in 2020, things regarding the epidemiology MDR gram-negative may have changed. Authors should at least provide data regarding any changes/measures they have taken since the time of the study.
2) Background: Authors could potentially include one paragraph with data regarding the epidemiology of MDR gram-negative infections in ICU and specifically NICU. This will assist the reader to understand the impact of the study
3) Methods: Authors may provide also information regarding the screening methods they use in NICU (MRSA, ESBL, CRE screening if any). Line 110 --> please change with early-onset.
4) Results: The authors should provide info regarding the antibiotic schemes that were used, especially the inadequate (monotherapy/combination therapy, which regimens, etc)
Lines 198-200: In the methods, section authors state that 7-day and 30-day mortality will be evaluated, thus the KM curve including deaths up to 120 d after is not consistent with the methods. Please revise accordingly.
5) Discussion: The authors should expand more and include future interventions in the NICU (antimicrobial stewardship and infection control) in order to reduce the inadequate therapy of MDR gram-negative infections in their NICU. Also, more data regarding the comparison of epidemiology of other NICU in Taiwan would be of interest.
Author Response
RE: Antibiotics-753672
Risk factors of initial inappropriate antibiotic therapy and the impacts on outcomes of neonates with gram-negative bacteremia
Dear Editor,
Thank you for your appreciated comments on our manuscript. We had the manuscript revised, all according to the reviewers’ and editor’s suggestions. We underline every change and highlight in red color on the revised manuscript. The replies for the reviewers’ criticisms are as followings. We hope this revised version can be acceptable.
Best regards,
Ming-Horng Tsai
Chief, Division of Neonatology and Pediatric Hematology/Oncology, Department of Pediatrics, Yunlin Chang Gung Memorial Hospital, Taiwan, R.O.C.
Comments from Reviewer No.3:

I read with interest the study. Overall it is an interesting study with important clinical implications that will serve as the baseline for the discussion for appropriate antimicrobial stewardship programs in Taiwan and also infection prevention and control measures to reduce the MDR gram-negative bacteria.
My comments can be found below.
1) I am concerned regarding the timeline of the study since the data are until Dec 2014. Since we are in 2020, things regarding the epidemiology MDR gram-negative may have changed. Authors should at least provide data regarding any changes/measures they have taken since the time of the study.
Reply:
Thank you for your instructive advice. I will provide data regarding any changes/measures they have taken since the time of the study. Since the time after the study, (Dec 2014), we started routine antifungal prophylaxis in our NICU since 2017, which resulted in changes of epidemiology and fungal species that caused neonatal candidemia in our NICU (this issue is mentioned in the discussion section, page 17, line 314-324). However, I did not notice that it caused any change in the epidemiology of GNB in our NICU. I think the epidemiology of MDR gram-negative bacteremia did not change a lot during the past 5 years, because our research team already had the data. I am writing another manuscript regarding the epidemiology of neonatal late-onset sepsis when compared with that in ten years ago (2004-2007). The data of ten years ago has been published in Pediatrics 2014;133(2):e322-9. (see reference no. 4, and I am the first author). However, because I can’t provide the data which I am going to submit to next journal, I think it quite difficult to provide any change/measure after the study period (after Dec 2014).
2) Background: Authors could potentially include one paragraph with data regarding the epidemiology of MDR gram-negative infections in ICU and specifically NICU. This will assist the reader to understand the impact of the study
Reply:
Thank you for your instructive advice. I will include one paragraph with data regarding the epidemiology of MDR gram-negative infections in ICU and specifically in NICU. In the middle of the 2nd paragraph of background, I add “MDR GNB accounts for nearly one-fifth of all neonatal GNB, or 7-10% of all neonatal late-onset sepsis [4,17,18]. Neonates with MDR GNB are well known to have a significantly higher risk of mortality than those with antibiotic-susceptible GNB [4,18].”
3) Methods: Authors may provide also information regarding the screening methods they use in NICU (MRSA, ESBL, CRE screening if any). Line 110 --> please change with early-onset.
Reply:
Thank you for your instructive advice. There is no screening method for MRSA, ESBL or CRE that we used in the NICU. I have mentioned the routine blood culture and microbiology that we used in our NICU as following in the Definition section of Methods (start from line 103)
“In our institution, blood cultures were incubated in the Bactec 9240 system, and antibiotic susceptibility testing was performed by using the disk diffusion method according to the recommendations of the Clinical Laboratory Standards Institute [20]. Multidrug resistance was defined using previously published criteria [21]. The microbiology laboratory would notify the clinician when a blood culture was positive, and the clinician was responsible to modify the antimicrobial regimens according to subsequent bacterial identification and antimicrobial susceptibility results.”
Also, the early-onset in line 110 is corrected in the revised manuscript, thank you.
4) Results: The authors should provide info regarding the antibiotic schemes that were used, especially the inadequate (monotherapy/combination therapy, which regimens, etc)
Lines 198-200: In the methods, section authors state that 7-day and 30-day mortality will be evaluated, thus the KM curve including deaths up to 120 d after is not consistent with the methods. Please revise accordingly.
Reply:
Thank you for your instructive advice. I will provide information regarding the antibiotic schemes that were used in our NICU in the definition section of the method sections, as following: (page 8, line 151-156)
In our NICU, empirical antibiotics were prescribed for the coverage of both Gram-positive and Gram-negative organisms, usually oxacillin or vancomycin plus cefotaxime or gentamicin, once late-onset sepsis was suspected. For early-onset sepsis, ampicillin plus gentamicin or cefotaxime were usually prescribed, depending on the physician’s decision.
For Lines 198-200, I will add the “Kaplan-Meier analysis was used to compare the survival situation of neonates with GNB treated with appropriate versus inappropriate antibiotics.“ in the method section, thank you.
5) Discussion: The authors should expand more and include future interventions in the NICU (antimicrobial stewardship and infection control) in order to reduce the inadequate therapy of MDR gram-negative infections in their NICU. Also, more data regarding the comparison of epidemiology of other NICU in Taiwan would be of interest.
Reply:
Thank you for your instructive advice. I will expand more and include future interventions in the NICU in order to reduce the inadequate therapy of MDR gram-negative infections in the NICU.
I will also provide data regarding the comparison of epidemiology of other NICU in Taiwan as following: (in the last two paragraph of the discussion section, middle of page 17, line 314-324)
This study highlighted the importance of further efforts to elucidate the optimal timing and aggressive antibiotic regimen for neonates who are at risk of antibiotic-resistant Gram-negative bacteremia. Antimicrobial stewardship interventions have been well studied to prompt appropriate antibiotics prescription in the NICU [37-39]. In our institute, routine antifungal prophylaxis for extremely preterm neonates has been launched since early 2017, and the influences as well as those of antimicrobial stewardship intervention are currently under investigation. Our previous studies found MDR GNB accounted for 18.6% of all neonatal GNB [4,7,16].Because colonization of the newborns with MDR Gram-negative pathogens makes them the potential source of nosocomial outbreaks [8,40], further routine surveillance of colonization is also suggested.

Round 2
Reviewer 3 Report
Dear Authors,
Thank you for addressing the majority of the comments and for improving the manuscript. However, I still do not understand the scientific reasoning behind the following: "Subsequent episodes of bacteremia in study patients within one month from the first episode of Gram-negative bacteremia were excluded due to better outcome analysis".
Moreover, statistical analysis does not exclude confounding.
Lastly, since authors state that they already have newer data regarding the GNB epidemiology, I think they should incorporate the data in this manuscript.
Author Response
RE: Antibiotics-753672
Risk factors of initial inappropriate antibiotic therapy and the impacts on outcomes of neonates with gram-negative bacteremia
Dear Editor,
Thank you for your appreciated comments on our manuscript. We had the manuscript revised, all according to the reviewers’ and editor’s suggestions. We underline every change and highlight in red color on the revised manuscript. The replies for the reviewers’ criticisms are as followings. We hope this revised version can be acceptable.
Best regards,
Ming-Horng Tsai
Chief, Division of Neonatology and Pediatric Hematology/Oncology, Department of Pediatrics, Yunlin Chang Gung Memorial Hospital, Taiwan, R.O.C.
Comments from Reviewer No.3:

Thank you for addressing the majority of the comments and for improving the manuscript. However, I still do not understand the scientific reasoning behind the following: “Subsequent episodes of bacteremia in study patients within one month from the first episode of Gram-negative bacteremia were excluded due to better outcome analysis.”
Reply:
Thank you for your question. Because the 7-day mortality and 30-day all cause mortality after the GNB were compared between neonates treated with appropriate and those treated with inappropriate antibiotics, subsequent episodes of bacteremia that happened within one month from the previous one in study patients would cause difficulties in considering event-attributable mortality if these episodes were not excluded.
2) Moreover, statistical analysis does not exclude confounding
Reply:
Thank you for your instructive advice. Because the aim of this study is to evaluate the risk factors of prescribing inappropriate antibiotics for neonates with GNB, this issue was examined by multivariate logistic regression analysis. The confounding factors have been excluded in this issue. I think the reviewer considered the second part: we did not use multivariate logistic regression to analyze the independent risk factors of final mortality. However, we just aimed to investigate the outcomes of neonates with GNB treated with inappropriate antibiotics when compared with those treated with appropriate antibiotics. Therefore, we did not use multivariate logistic regression. For the independent risk factors of final mortality in neonates with GNB, this issue has been published in another study of our research team (see reference no.4).
3) Lastly, since authors state that they already have newer data regarding the GNB epidemiology, I think they should incorporate the data in this manuscript
Reply:
Thank you for your instructive advice. During the past ten years, we have started routine antifungal prophylaxis in our NICU, which caused a significant increase of non-albicans candidemia in our NICU. However, the effects on the epidemiology of GNB have not been well investigated. We are currently prompt antimicrobial stewardship program in our NICU. However, the results of the epidemiology of GNB are only preliminary, and not consolidated enough to be incorporated into this manuscript. I suggest not extending the issue in this manuscript, to avoid confusing and insignificant understanding, thank you.

Round 3
Reviewer 3 Report
Dear Authors
Thank you for the revised version of the manuscript. I would suggest revising the sentence "the first episode of Gram-negative bacteremia were excluded due to better outcome analysis" in order not to create misunderstanding from the readers.